# Relationship between Sleep Duration and Osteoarthritis in Middle-Aged and Older Women: A Nationwide Population-Based Study

**DOI:** 10.3390/jcm8030356

**Published:** 2019-03-13

**Authors:** Hye-Min Park, Yu-Jin Kwon, Hyoung-Sik Kim, Yong-Jae Lee

**Affiliations:** 1Department of Family Medicine, Yonsei University College of Medicine, 211 Eonju-ro, Gangnam-gu, Seoul 06273, Korea; jadore-hm@yuhs.ac (H.-M.P.); digda3@yuhs.ac (Y.-J.K.); 2Department of Medicine, Graduate School, Yonsei University, 50-1 Yonsei-ro, Seodaemoon-gu, Seoul 03722, Korea; 3Department of Family Medicine, Yonsei University College of Medicine, 50-1 Yonsei-ro, Seodaemoon-gu, Seoul 06273, Korea; ysos111@yuhs.ac

**Keywords:** sleep duration, osteoarthritis, KNHANES

## Abstract

(1) Background: Both long and short sleep durations have been associated with negative health outcomes, particularly in middle-aged and older adults. To date, there has been little research on the association between sleep and osteoarthritis. This study aimed to evaluate the relationship between sleep duration and radiographically confirmed osteoarthritis in middle-aged and older women. (2) Methods: This study included 5268 women aged ≥50 years from the Korea National Health and Nutrition Examination Survey. Sleep duration was categorized into four groups (≤5 h, 6 h, 7–8 h, and ≥9 h) using responses from a self-reported questionnaire, and 7–8 h was set as an appropriate sleep duration. Osteoarthritis was defined as Kellgren–Lawrence grade ≥2 in the knee or hip area in radiographic images with knee or hip joint pain. The odds ratios (ORs) and 95% confidence intervals (CIs) of osteoarthritis according to sleep duration were calculated using multiple logistic regression analyses. (3) Results: The prevalence of osteoarthritis according to sleep duration showed a U-shaped curve, with the nadir in the appropriate sleep category (7–8 h). Compared with the 7–8 h sleep duration, the ORs (95% CIs) of osteoarthritis in the short sleep duration (≤5 h/day) and long sleep duration (≥9 h/day) were 1.343 (1.072–1.682) and 1.388 (1.020–1.889), respectively, after adjusting for age, body mass index, current smoking, alcohol consumption, regular exercise, occupation, residential area, hypertension, type 2 diabetes, cardiovascular disease, and stroke. (4) Conclusions: Short and long sleep duration were positively associated with osteoarthritis in middle-aged and older women.

## 1. Introduction

Osteoarthritis is characterized by gradual loss of articular cartilage and secondary subchondral bone changes, which lead to pain, swelling, and osteophyte formation, particularly in the weight bearing joints [1]. Osteoarthritis is the leading cause of physical disability and is associated with increased health care utilization and costs and impaired quality of life in middle-aged and older populations [2].

Both long and short sleep durations have been associated with negative health outcomes, particularly in middle-aged and older adults. Inadequate sleep duration is known to contribute to increase risk of obesity, metabolic syndrome, type 2 diabetes, and cardiovascular disease [3,4,5]. Moreover, getting less than or more than optimal sleep duration of 7–8 h has been associated with increased levels of inflammatory markers such as C-reactive protein (CRP) and interleukin-6 (IL-6) [6,7]. Osteoarthritis was thought in the past to be a consequence of the aging process, and was thus termed as a degenerative joint disease. However, osteoarthritis is now considered to result from a multi-factorial pathogenesis, including genetic predisposition, mechanical forces, and a low-grade inflammation associated with the advanced stages of cartilage degeneration and a complex cellular and biochemical process of chondrocyte [8,9].

Of the daily lifestyle factors, physical activity plays a key role in the development and progression of osteoarthritis. Moderate physical activity has been shown to enhance chondroprotective glycoproteins such as lubricin in aging animal models [10]. Moreover, nutritional imbalance such as heavy consumption of junk food or animal fat and insufficient intake of vitamins C and D are closely linked to chronic inflammation, and are thereby involved in the development of osteoarthritis [11]. The supplementation of extra-virgin olive oil with physical activity increased the expression of lubricin in synovial fluid of rats [12]. To date, little has been investigated about the relationship between sleep duration, another crucial lifestyle factor, and osteoarthritis. Therefore, this study aimed to examine the association between sleep duration and radiologically confirmed knee or hip osteoarthritis in a representative sample of middle-aged and older women.

## 2. Experimental Section

### 2.1. Data Collection

This study was based on data obtained from the Fifth Korea National Health and Nutrition Examination Survey (KNHANES-V), a nationally representative survey conducted by the Korea Centers for Disease Control and Prevention between 2010 and 2012. The target population of this survey was noninstitutionalized Korean women. The sampling units were households selected through a stratified, multistage, probability-sampling design that was based on geographic area, sex, and age group using household registries. To produce results representative of the entire Korean population, sampling weights indicating the probability of being sampled were assigned to each participant. Participants answered a four-part questionnaire that included a health interview, a health behavior survey, a health examination, and a nutrition survey. For the 2010–2012 KNHANES, citizens were informed that they had been randomly selected as a household to voluntarily participate in the nationally representative survey conducted by the Korean Ministry of Health and Welfare, and that they had the right to refuse to participate in accordance with the National Health Enhancement Act supported by the National Statistics Law of Korea. All study participants provided written informed consent. 

As radiographic images of the knee and hip were only taken in subjects aged ≥50 years, a total of 5724 participants were eligible for this study. Of them, 456 participants with missing data were excluded and 5268 women were included in the final analysis. The KNHANES was approved by the Institutional Review Board of the Korea Centers for Disease Control and Prevention (IRB No. 2010-02CON-21-C, 2011-02CON-06-C, 2012-01EXP-01-2C). 

Physical examinations were performed by trained medical staff. Body mass index (BMI) was calculated as the proportion of weight (kg) to height^2^ (m^2^). Blood pressure (BP) was measured in the right arm using a standard mercury sphygmomanometer (Baumanometer, Copiague, NY, USA). The heath interview collected data on the participant’s age, sex, residential area, occupation, and health-related behavior (exercise, cigarette smoking, and alcohol consumption) by a self-reported questionnaire. Smoking status was categorized as a non-smoker, ex-smoker, and current smoker. Alcohol consumption was categorized into two groups based on how frequently the participant consumed any type of alcohol; current drinkers were defined as those who had consumed alcohol ≥2–3 times a week for the past year. Regular exercise was regarded as walking ≥3 times per week for ≥30 min at a time. Seep duration information consisted of self-reported data obtained by response to the question, ‘How many hours do you usually sleep a day?’. The responses were classified into four categories (≤5 h, 6 h, 7–8 h, and ≥9 h). On the basis of the sleep definitions of the National Sleep Foundation, 7–8 h was set as an appropriate sleep duration in our study [13]. Occupations were classified into the following four groups based on a previous study: white-collar (WC) workers, encompassing professionals, office employees, and managers; pink-collar (PC) workers, containing service and sales staff; blue-collar (BC) workers, including machinists and technical engineers; and agribusiness and low-level (AL) workers, consisting of agriculture and fish farm expert workers and low-class laborers. Residential area was categorized as urban or rural based on the administrative division. Hypertension (HTN), type 2 diabetes, cardiovascular disease (CVD), and stroke were defined as previous diagnosis by a physician. 

### 2.2. Definition of Osteoarthritis 

Radiographic examinations of the knee and hip were performed using a SD 3000 Synchro Stand instrument (Accele Ray, Bern, Switzerland). All radiographic images were reviewed by two radiologists using the Kellgren–Lawrence grading system. Weight-bearing anteroposterior, bilateral anteroposterior, and lateral plain radiographs were measured to evaluate the knees. Bilateral and anteroposterior plain radiographs were obtained to examine the hip. The degree of radiographic osteoarthritis was assessed according to the KL grading system as follows: Grade 0, none: no visible features of osteoarthritis; Grade 1, doubtful: questionable osteophytes or questionable joint space narrowing; Grade 2, minimal: definitive small osteophytes, minimal/mild joint space narrowing; Grade 3, moderate: definitive moderate osteophytes, joint space narrowing of at least 50%; and Grade 4, severe: severely impaired joint space, subchondral bone cysts, and sclerosis [14]. In this study, radiographic osteoarthritis was defined as KL grade ≥2 on either the knee or hip area in radiographic images [15]. All participants also reported whether they had suffered symptoms in the hip and knees for more than 30 days in the past three months. A person who met the two following criteria was defined as having osteoarthritis: (1) hip and knee pain lasting for more than 30 days in the past three months and (2) KL grade ≥2 on either the knee or hip area in the radiographic images.

### 2.3. Statistical Analysis

All data are presented as mean ± standard error (SE) or percentage ± SE based on a stratified, multistage, and probability-sampling design. Differences in characteristics across osteoarthritis categories were summarized using the weighted one-way analysis of variance (ANOVA) for continuous variables and the weighted chi-square test for categorical variables. The proportion between-group difference was also assessed by post hoc analysis of weighted chi-square test.

The odds ratios (ORs) and 95% confidence intervals (CIs) for the presence of osteoarthritis according to sleep duration category were calculated using multiple logistic regression analyses after adjusting for potential confounding variables. All analyses were performed using SPSS statistical software, version 23.0 (Version 23.0; IBM Corp., Armonk, NY, USA). All statistical tests were two-sided, and statistical significance was defined as a *p* < 0.05.

## 3. Results

### 3.1. Study Population and Baseline Characteristics

Table 1 shows the demographic and clinical characteristics of the study population according to presence of osteoarthritis. The mean age was 63.3 ± 0.2 years and the prevalence of osteoarthritis was 20.9%. The proportion of regular exercise was lower and the participants who lived in a rural area had a higher prevalence of osteoarthritis. The prevalence of hypertension, type 2 diabetes, CVD, and stroke was more prevalent in individuals with osteoarthritis.

### 3.2. Relationship between Sleep Duration and Osteoarthritis

Figure 1 shows the prevalence of radiographically confirmed osteoarthritis according to four categories of sleep duration (≤5 h, 6 h, 7–8 h, and ≥9 h). There was a U-shaped curve with the nadir in the appropriate sleep category (7–8 h): 25.4%, 17.3%, 15.5%, and 26.1%.

Table 2 shows the results of multiple logistic regression analyses to assess the association between sleep duration and osteoarthritis. Compared with the 7–8 h of sleep duration, the ORs (95% CIs) of osteoarthritis in the short sleep duration (≤5 h/day) and the long sleep duration (≥9 h/day) were 1.343 (1.072–1.682) and 1.388 (1.020–1.889), respectively, after adjusting for age, body mass index, current smoking, alcohol consumption, regular exercise, occupation, residual area, hypertension, type 2 diabetes, CVD, and stroke. 

## 4. Discussion

The present study identified U-shaped associations between short and long sleep duration and osteoarthritis in middle-aged and older women. Our findings are in agreement with the previous results that both long and short sleep durations have been associated with negative health outcomes in middle-aged and older adults. Previous studies have linked short or long sleep duration with an increased risk for all-cause mortality, as well as obesity, hypertension, and type 2 diabetes from cross-sectional and longitudinal studies [3,4,5].

Although the exact mechanism underlying the observed association between sleep duration and osteoarthritis is uncertain, some explanations may be offered. First, short and long sleep duration and osteoarthritis could be linked to reactive oxidative stress and low-grade inflammation, which in turn may induce chondrocyte inflammatory response. Recent evidence suggests that reactive oxygen species have an important role in the development of osteoarthritis [16], and sleep deprivation or short sleep lead to increased oxidative stress and inflammatory markers in human and animal studies [6,17,18]. Long sleep duration also leads to increased levels of inflammatory markers such as CRP, and pro-inflammatory cytokines such as interleukin (IL)-6, and tumor necrosis factor (TNF)-α [6,19]. Emerging evidence suggests that low-grade inflammation is closely related with the initiation and progression of osteoarthritis. Pro-inflammatory cytokines such as IL-6 and TNF-α levels are elevated in the synovial fluid and cartilage of patients with osteoarthritis and chronic elevations in cytokines are associated with an increased risk of osteoarthritis [20,21]. These pro-inflammatory cytokines upregulate inflammatory response and inhibit the synthesis of proteoglycan and type II collagen in chondrocytes [21]. Second, both short and long sleep are well established risk factors of future body weight and fat gain, which may contribute to the increased risk of osteoarthritis [19,20,21]. In a study of 276 Canadian adults from the Quebec Family Cohort Study, the incidence risk of obesity was 1.27 times higher for short duration sleepers and 1.21 times higher for long duration sleepers when compared with average-duration sleepers [22]. Thus, the cascade from extreme sleep duration to body weight gain may contribute to the pathogenesis of osteoarthritis. This study should be taken into consideration when interpreting the results of the present study. This was a cross-sectional study and was unable to establish a causal relationship between sleep duration and osteoarthritis. The results might reflect reverse causality and a bidirectional relationship in the association between sleep duration and osteoarthritis. Therefore, further prospective studies with long follow-up duration are warranted to determine whether there is a cause-and-effect relationship between sleep duration and development of osteoarthritis. Moreover, because we used secondary data from KNHANES, we could not evaluate sleep quality and plasma or synovial inflammatory markers such as IL-6 and TNF-α. Despite these potential limitations, our results can be generalized to the entire Korean population by applying complex sampling design analysis. In addition, we used an objective tool to detect osteoarthritis in the knee and hip joints based on objective radiological findings as well as osteoarthritis-related symptoms.

## 5. Conclusions

In conclusion, we found U-shaped associations between short and long sleep duration and osteoarthritis in middle-aged and older women. This study implies that sleep duration could be at least partly associated with osteoarthritis. Our preliminary and hypothetical results are based on the fact that OA is now being regarded as a multifactorial disease and future research is needed to confirm this hypothesis. Considering sleep education and sleep duration could be additional useful lifestyle guidelines when treating osteoarthritis patients.

## Figures and Tables

**Figure 1 jcm-08-00356-f001:**
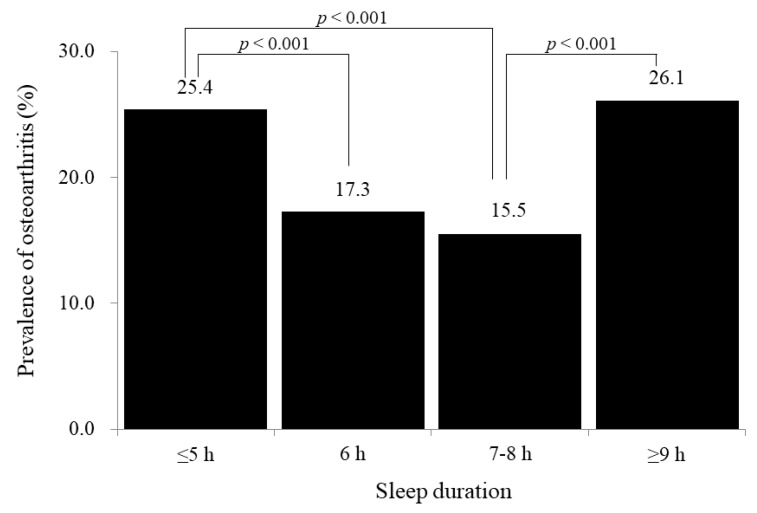
Prevalence of osteoarthritis according to sleep duration (*p*-values were calculated by post hoc analysis of weighted chi-square test between-groups).

**Table 1 jcm-08-00356-t001:** Clinical characteristics in the study population.

	Radiologically Confirmed Osteoarthritis		
	Presence	Absence	*p* Value
Unweighted *N*	1063	4205	
Age (years)	69.6 ± 0.4	61.40 ± 0.2	<0.001
Body mass index (kg/m^2^)	25.3 ± 0.1	24.0 ± 0.1	<0.001
Current smoking (%)	4.4 ± 0.7	4.8 ± 0.4	0.718
Alcohol drinking (%)	5.0 ± 0.9	6.2 ± 0.4	0.290
Regular exercise (%)	54.0 ± 2.2	61.9 ± 1.2	<0.001
Occupation (%)			<0.001
Agriculture	24.2 ± 2.0	23.8 ± 1.1	
Blue collar	1.4 ± 0.4	2.6 ± 0.3	
White collar	0.8 ± 0.3	4.5 ± 0.4	
Pink collar	8.4 ± 1.1	15.8 ± 0.7	
Unemployment	65.1 ± 2.2	53.3 ± 1.1	
Residential area (%)			<0.001
Urban area	61.0 ± 2.9	74.7 ± 2.0	
Rural area	39.0 ± 2.9	25.3 ± 2.0	
Hypertension (%)	57.3 ± 1.9	36.9 ± 1.0	<0.001
Type 2 diabetes (%)	18.9 ± 1.7	11.4 ± 0.6	<0.001
Cardiovascular disease (%)	6.5 ± 0.8	4.3 ± 0.4	0.014
Stroke (%)	4.0 ± 0.8	2.6 ± 0.3	0.049

**Table 2 jcm-08-00356-t002:** Odds ratios and 95% confidence intervals for the presence of osteoarthritis according to sleep duration.

	Sleep Duration			
	≤5 h	6 h	7–8 h	≥9 h
Model 1	1.861 (1.523–2.273)	1.139 (0.900–1.442)	1.000 (reference)	1.927 (1.412–2.629)
Model 2	1.357 (1.089–1.690)	1.079 (0.840–1.386)	1.000 (reference)	1.391 (0.999–1.936)
Model 3	1.343 (1.072–1.682)	1.092 (0.836–1.427)	1.000 (reference)	1.388 (1.020–1.889)

Model 1: unadjusted. Model 2: adjusted for age and body mass index. Model 3: adjusted for age, body mass index, current smoking, alcohol drinking, regular exercise, occupation, residential area, hypertension, type 2 diabetes, cardiovascular disease, and stroke.

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
