# Peer review of "Relationship between Sleep Duration and Osteoarthritis in Middle-Aged and Older Women: A Nationwide Population-Based Study"

_jcm, 2019, doi:10.3390/jcm8030356_

Round 1
Reviewer 1 Report
Manuscript titled “Relationship between sleep duration and osteoarthritis in middle-aged and older women: A nationwide population-based study” deals an important issue of medical cartilage biology. This study aimed to evaluate the relationship between sleep duration and radiographically confirmed osteoarthritis in middle-aged and older women. The authors concluded that short and long sleep duration was positively associated with osteoarthritis in middle-aged and older women.
The work is good, well written, fluent and interesting, moreover, the state of the art/background is very poor and miss some important information.
Please strengthen and update better the introduction section adding more details and interesting information about the correlation lifestyle (exercise, aging, nutrition) and OA. I recommend to see the following recent and interesting papers or others and comment them to stay to the study topic:
Osteoarthritis in the XXIst century: risk factors and behaviours that influence disease onset and progression. Int J Mol Sci. 2015 Mar 16;16(3):6093-112. Extra-virgin olive oil diet and mild physical activity prevent cartilage degeneration in an osteoarthritis model. An "in vivo" and "in vitro" study on lubricin expression. Journal of Nutritional Biochemistry 2013, 24; 2064-2075.
Physical activity ameliorates cartilage degeneration in a rat model of aging: A study on lubricin expression. Scandinavian Journal of Medicine & Science in Sports. 2015: 25: e222–e230.
In the conclusion please specify the clinical relevance of your work and please stated that these are preliminary and hypothetical results due to the fact that OA is a multifactorial disease, so further studies are needed to confirm the present ones.
Author Response
Reply to the Reviewer # 1:
Manuscript titled “Relationship between sleep duration and osteoarthritis in middle-aged and older women: A nationwide population-based study” deals an important issue of medical cartilage biology. This study aimed to evaluate the relationship between sleep duration and radiographically confirmed osteoarthritis in middle-aged and older women. The authors concluded that short and long sleep duration was positively associated with osteoarthritis in middle-aged and older women. The work is good, well written, fluent and interesting, moreover, the state of the art/background is very poor and miss some important information.
1. Please strengthen and update better the introduction section adding more details and interesting information about the correlation lifestyle (exercise, aging, nutrition) and OA. I recommend to see the following recent and interesting papers or others and comment them to stay to the study topic:
Osteoarthritis in the XXIst century: risk factors and behaviours that influence disease onset and progression. Int J Mol Sci. 2015 Mar 16;16(3):6093-112.
Extra-virgin olive oil diet and mild physical activity prevent cartilage degeneration in an osteoarthritis model. An “in vivo” and “in vitro” study on lubricin expression. Journal of Nutritional Biochemistry 2013, 24; 2064-2075.
Physical activity ameliorates cartilage degeneration in a rat model of aging: A study on lubricin expression. Scandinavian Journal of Medicine & Science in Sports. 2015: 25: e222–e230.
Response: We really appreciate your valuable suggestions. We have added the following sentences in the Introduction section of the revised manuscript as follows:
“Of daily lifestyle factors, physical activity plays a key role in the development and progression of OA. Moderate physical activity has been shown to enhance chondroprotective glycoproteins such as lubricin in aging animal models. [14] Moreover, nutritional imbalance such as heavy consumption of junk food or animal fat and insufficient intake of vitamins C and D are closely linked to chronic inflammation, thereby involved in the development of OA. [12] The supplementation of extra-virgin olive oil with physical activity increased the expression of lubricin in synovial fluid of rats. [13].”
2. In the conclusion please specify the clinical relevance of your work and please stated that these are preliminary and hypothetical results due to the fact that OA is a multifactorial disease, so further studies are needed to confirm the present ones.
Response: Thank you so much for your insightful comments. In accordance with the reviewer’s recommendations, we have added the following sentences in the Discussion section of the revised manuscript: “This study implicates sleep duration could be at least partly associated with osteoarthritis. Our preliminary and hypothetical results are based on the fact that OA is now being regarded as a multifactorial disease and future research is needed to confirm this hypothesis. Considering sleep education and sleep duration could be additional useful lifestyle guideline when treating osteoarthritis patients.”
Reviewer 2 Report
The authors present an analysis of a large cohort of middle-aged women to study the association between osteoarthritis and sleep duration. The findings are reported in a clear and logical manner. I have only a couple of comments about how the manuscript could be improved.
First, this study examines the relationship of sleep duration and osteoarthritis, but several times the authors mention circadian dysregulation as a potential contributing factor (e.g., lines 52-54). However, they do not have data supporting a role of the circadian clock in their observations, and it is important to keep in mind that sleep duration is independent from circadian dysregulation (even though the sleep and circadian system are closely related). It is suggested that the authors remove reference to circadian dysregulation from their Introduction, as this information is not relevant to the results.
What analysis was conducted to generate the p-value in Figure 1? The figure as currently drawn indicates p<0.001 for both the shortest and longest sleep duration, and it is presumed that 7-8 h of sleep was used as the reference group in post-hoc pairwise comparisons, but this should be more explicitly stated and the figure redrawn to reflect the actual comparisons that were statistically significant.
Author Response
Reply to the Reviewer # 2:
The authors present an analysis of a large cohort of middle-aged women to study the association between osteoarthritis and sleep duration. The findings are reported in a clear and logical manner. I have only a couple of comments about how the manuscript could be improved.
1. First, this study examines the relationship of sleep duration and osteoarthritis, but several times the authors mention circadian dysregulation as a potential contributing factor (e.g., lines 52-54). However, they do not have data supporting a role of the circadian clock in their observations, and it is important to keep in mind that sleep duration is independent from circadian dysregulation (even though the sleep and circadian system are closely related). It is suggested that the authors remove reference to circadian dysregulation from their Introduction, as this information is not relevant to the results.
Response: We totally agree with the reviewer’s comment. As recommended by the reviewer, we have deleted the following sentence from the initial manuscript: “A recent animal study found that circadian clock genes in chondrocytes regulate cartilage homeostasis, supporting a role of circadian dysregulation in the pathogenesis of osteoarthritis.”
2. What analysis was conducted to generate the p-value in Figure 1? The figure as currently drawn indicates p<0.001 for both the shortest and longest sleep duration, and it is presumed that 7-8 h of sleep was used as the reference group in post-hoc pair wise comparisons, but this should be more explicitly stated and the figure redrawn to reflect the actual comparisons that were statistically significant. This is an excellent article and well researched and presented.
Response: We really appreciate the reviewer’s insightful and helpful suggestions. We have presented the p-values in the revised Figure 1 with revised legends “Prevalence of osteoarthritis according to sleep duration (p values were calculated by post hoc analysis of weighted chi-square test between-groups)